# Development and validation of an instrument to measure personality in adolescence: The HEXACO Medium School Inventory Extended (MSI-E)

**Augusto Gnisci**[1]*, **Francesca Mottola**[1], **Marco Perugini**[2], **Vincenzo Paolo Senese**[1], **Ida Sergi**[1]

1 Department of Psychology, University of Campania "Luigi Vanvitelli", Caserta, Italy, 2 Department of Psychology, University of Milan-Bicocca, Milan, Italy

* augusto.gnisci@unicampania.it

**Data Availability Statement:** All the used data files will be available from the OSFHOME database

## Abstract

In this paper, we aimed at developing and validating a novel instrument to evaluate personality in 10–14 years old adolescents with six basic traits, with two dedicated studies. In Study 1, we generated a large pool of items (384 items) from three basic items sources, which we administered to 714 Italian adolescents. Using principal component analysis (PCA) and extension factor analysis, we selected the best eight items for each facet, and so the best 32 items for each factor, except for the Unconventionality facet of Openness to Experience (O) for which we selected the best six items. This resulted in a total of 190 items. The 190-item HEXACO-MSI had very good levels of dimensional validity and reliability, but it fell short in containing 8 items for each facet (i.e., for Unconventionality) and in balancing normal and reversed items within each facet. Therefore, in a second study we added items to the scale and verified again the dimensionality and reliability with the goal of developing a final version of the scale. In Study 2, we administered a version of the HEXACO-MSI consisting of 219 items to 1175 Italian adolescents. Using principal component analysis (PCA), we selected the best eight items for each facet equally balanced between normal and reversed items within each facet and factor. Confirmatory factor analysis (CFA) confirmed the six-factor structure and its invariance. The results showed that the HEXACO-MSI-E had a clear six-factor structure in adolescents, that was invariant across gender and across the three middle school classes, and was reliable. Finally, we established temporal stability of each factor in two measurements after one year. Together with the positive results of this contribution, we discussed some aspects for future studies.

## Introduction

It has been over 40 years since most personality researchers came to agree that the domain of personality variation could be best summarized in terms of few personality traits [1].

(https://osf.io) (accession number DOI 10.17605/OSF.IO/NATQC).

**Funding:** The project EMERGE (ID 389) has been funded by the programme V:ALERE 2019 of the University of Campania "Luigi Vanvitelli" (D.R. 906 del 4/10/2019, prot. n. 157264, 17/10/2019). The funders had no role in study design, data collection and analysis, decision to publish, or preparation of the manuscript.

**Competing interests:** The authors have declared that no competing interests exist.

Personality trait measures are constructed using a lexical strategy in which ratings of personality-descriptive adjectives are subjected to factor analysis to determine the structure of personality [2]. The Big Five personality traits, also known as the Five Factor Model (FFM), is based on these types of language descriptors of personality. These five factors are labeled: Extraversion, Agreeableness, Conscientiousness, Neuroticism and Openness. In the last 15 years, despite many lexical studies providing support for the Big Five, several more lexical studies have provided support for an alternative personality taxonomy [2, 3]. This alternative structure is based on the same lexical and cross-cultural studies from which originated the FFM, but it is composed of six dimensions instead of five [4]. The names of the six dimensions (whose acronym is HEXACO) are: Honesty–Humility (H), Emotionality (E), eXtraversion (X), Agreeableness (A), Conscientiousness (C) and Openness to Experience (O). The six-factor structure partly overlaps with the classic Big Five. In fact, eXtraversion, Conscientiousness, and Openness to Experience largely correspond to their counterparts in the Big Five, except for the exclusion of the facet intellectual ability from HEXACO Openness to Experience. Otherwise, the HEXACO model differs from the FFM model in terms of redefining Agreeableness and Neuroticism, and adding the Honesty-Humility factor [2]. These three factors contain the variance associated with the Big Five factors Agreeableness and Neuroticism as also the additional variance not captured in the classic Big Five. To use Goldberg's [5] term, Big Five variance is "reorganized" into the six-factor structure. In the HEXACO model, adjectives that typically define the Emotionality factor include vulnerable, sensitive, anxious, and sentimental. This factor differs from the FFM's Neuroticism, in particular it does not include the aspects associated with anger. Adjectives that typically define the Agreeableness factor include peaceful, meek, patient, and agreeable and in the opposite pole irascible, choleric, stubborn, and argumentative. This factor shares some similarities with the classic Big Five Agreeableness (e.g., meekness), but lacks the aspects associated with sentimentality, while contains (at its negative pole) the traits associated with the Big Five Neuroticism. Adjectives that typically define Honesty-Humility include honest, sincere, fair, and modest and in the opposite pole greedy, conceited, deceitful, and pretentious. Honesty-Humility was only modestly related to the FFM factors, but it often showed modest associations with FFM Agreeableness domain. This relation was mainly due to the Straightforwardness and Modesty facets of FFM Agreeableness [2].

Many studies [2, 6] have shown that the six-dimensional space captures some important personality variations not represented in the five-dimensional model, while also improving the theoretical interpretation of personality variation. Indeed, within educational and adolescent contexts, the Honesty-Humility factor was found to be a more valid predictor of bullying [7–9], proactive aggression [10–12], relational aggression [13], antisocial behaviors at school [14] and counterproductive academic behaviors [15], and it was also found to be a better predictor of student outcomes, such as student grades and prosocial academic behaviors [15–17]. All the results just mentioned showed that HEXACO model gives unique and meaningful contributions to the study of children and adolescents' behaviors.

An important aspect to be addressed for the assessment of traits in children and adolescents is the use of self-report personality measures [18]. Criticism has been addressed in particular to the adequate understanding of the questions of standard adult personality inventories and/or ability to answer questions about oneself. The topics covered in these standard questions are often not relevant for a child/adolescent. On the basis of these criticisms, it is clear that adult personality inventories are necessarily inappropriate for children as young as 12 years, therefore simplified inventories have been developed [18–20]. Most studies carried out on children used the Five Factor Model as theoretical framework and one of the most used scale was the Big Five Questionnaire-Children (BFQ-C) [20, 21]. Only recently, some authors [22] have adopted the HEXACO model for investigating personality in adolescence (age 11–17 years),

supporting the potential advantages of this model compared to Big Five model. However, the authors used the adult version of HEXACO scale rather than adapting the content of the items to the adolescents' age. Recently, Sergi and colleagues have developed and validated a preliminary version of a HEXACO scale adapted for 10–14 years old adolescents named HEXACO-Middle School Inventory (HEXACO-MSI) [16]. This scale has been shown to have adequate psychometric properties and provided important results about many aspects of validity, for example, three factors, Conscientiousness, Honesty-Humility, and eXtraversion, proved good predictors of school grades. Despite several merits, this preliminary version of the scale also presented some limitations. In fact, in validating the instrument, the items were selected on the basis of the strength of their unique loadings on the pertinent dimension. This strategy allowed to choose the best items for the pertinent dimension but did not allow to respect other important elements, such as to represent all the facets within each dimension, to present a balanced number of positive and negative (i.e., reversed) items within each facet and, by consequence, each factor, and to have a sufficiently high number of items in each facet in order to improve the representativeness of the facet and the factors. In brief, the first HEXACO-MSI had some weaknesses regarding content validity.

Last, but not least, an important point to evaluate when considering an age group such as adolescence is the stability of traits. The results of previous research showed that personality traits are moderately stable in preschool years, become increasingly stable until middle adulthood [23–25] and the mean levels of personality traits among adolescents resemble the respective scores of the adult population [26, 27]. These data refer to studies with the FFM. Much of the information regarding the association between age and personality dimensions in the HEXACO model is due to the cross-sectional study of Ashton & Lee [28]. The results indicate similarities with previous findings based on the Big Five.

Only few studies, however, have used the HEXACO model of personality among adolescents and only Sergi and colleagues using the preliminary version of HEXACO-MSI [16] verified test-retest reliability, which was good but measured at a relatively short distance of one month.

Thus, the general aim of this contribution is to develop a new extended version of the HEXACO-Medium School Inventory (HEXACO-MSI-E) for adolescents with improved psychometric qualities (i.e., increased number of items, representativeness of the items with respect to dimensions and facets, equilibrium between normal and reversed items). This will be realized with two studies. Within the general aim, specific aims were: (a) to provide a wide sample of items whose domain is representative, in adolescence, of all the contents, the facets and the dimensions of the HEXACO model, possibly balancing normal and reversed items (content validity); (b) to verify the six-dimensional structure of personality in adolescence–that is, the loading of each item on the expected dimension, the absence of cross-loading on not pertinent factors, and low or modest correlations between traits–in different samples of adolescents. This will be done by exploratory and confirmatory factorial analysis techniques on both items and facets (dimensional validity); (c) to provide evidence that this structure of personality is invariant in males and females and in the three classes of medium school (7th, 8th and 9th grade) (cross-validation); (d) to provide repeated evidence that traits of the adolescents are reliable by Cronbach's alpha and omega coefficients (internal consistency); (e) to proving evidence of the temporal stability of the trait in adolescents after a relatively long time distance of one year (test-retest reliability).

## Study 1

Study 1 was conducted to develop a new version of the questionnaire, the HEXACO Middle School Inventory Extended (HEXACO-MSI-E) to measure personality traits in adolescents

(10–14 years old) using the HEXACO as a theoretical model of reference. The aim was to develop an instrument that would show good psychometric properties when administered to large samples of students from middle school, and represent both a quantitative (i.e., more items, facet-level measures) and a qualitative (i.e., better representation of facets, more balanced items) improvement over the previous HEXACO-MSI by Sergi and colleagues [16]. Specifically, once identified a large number of items representative of the facets of the inventory, we wanted to select the best eight items for each facet, establish internal consistency reliability, low inter-scale correlations, and a factor structure in which items (or facets) of the same scale show their primary loadings on the same factor of the six-factor solution. In light of these considerations, first, a large pool of items was generated, and then it was administered to a large sample of Italian adolescents to evaluate their psychometric characteristics.

## Methods

**Participants and procedure.** Participants were 714 middle school students (52.7% Females, Mean age = 11.94, SD = 0.91) recruited form seven schools of Campania (Italy), 47.8% attending the 6th, 34.6% the 7th and 17.6% the 8th grade. For each child, a parent or legal guardian was also involved to get indirect information about students' behaviors.

Recruitment procedure and informed consent. First, the project received the approval by the local Ethics Committee of the Department of the first author (approval number 13/26.05.2020). Then, the research plan was approved by the Directors of the schools and by their Council, that culminated in a formal informed consensus, signed by the Directors. Third, parents and adolescents were informed about the project by the school, by the research assistant, by written instructions and by video- and audio-recordings, specially prepared. Fourth, once duly informed, the parents/legal guardians were administered (online) the protocol, in the beginning of which they read the basic information regarding the research and then provided, if they wanted, the authorization to the participation for their children and then for themselves. Fifth, at the beginning of their online protocol, the adolescents read a written description of the research and were asked their willingness to participate to the research. It was specified that the responses were recorded in an anonymous way and data were treated collectively. The protocol filled out by parents and their children were associated through an alphanumeric code generated by each participant on the basis of general questions to guarantee anonymity. The children and their parents were, then, informed about the project and that they were free to decline to take part in the data collection at any time they wished and without any consequence.

Administration. Data were collected in May and June 2020, right after the so-called first lockdown imposed by the government in Italy for the COVID-19 pandemic. Participants completed online protocol on Qualtrics platform. According to the ministerial indications, the involved schools provided distance education, using online platforms, with the activation of virtual classrooms to guarantee continuity of learning for students. For this reason, children were administered the online protocol in their virtual classrooms by research assistants and at the presence of the teacher. Considering the high number of items, the students completed it in two sessions on average 1.4 (SD = 0.69) days apart (97.8% ≤ 2 days). The involved parents were previously contacted and instructed by research assistants, who, also, sent them the link to the protocol. They were asked to fill out the online protocol in a few days. It was also specified that only one parent or caregiver was asked to complete the protocol.

**Measures.** *Demographic information*. At the beginning of the protocol, basic information such as gender, age, class, were requested.

*Initial pool of items for developing the HEXACO-MSI-E*. To identify the items to be administered to adolescents, we have considered three important sources: (a) the 96-item pool from

which was extracted the final version of the 48-item HEXACO-MSI in the preceding study [16]; (b) the International Personality Item Pool (IPIP) [29]; (3) the HEXACO Simplified Personality Inventory (HEXACO-SPI) [30]. We identified the items that we regarded as best to be adapted for adolescents: 95 items from the HEXACO-MSI, 190 from the IPIP and 90 from the HEXACO-SPI. To balance the items within each facet and to have the same items for each of them, we selected 3 additional items from the 192-item HEXACO-PI-R [31] and developed six more items. It resulted in a total of 384 items (64 for each factor, 16 items for facet; in 8 facets the ratio between normal and reversed items was 8:8, in 9 was 7:9, in 6 was 9:7, in only one case was 6:10), that constituted the initial pool. The items in English were translated in Italian by a linguist and her translation supervised by three psychologists expert on personality. Then, when considered necessary the item stem was simplified and adapted to adolescence. Therefore, the pool of item (384 items) was administered to adolescents. For each item, responses were collected using a 5-step Likert-type scale: from 1 (True) to 5 (False), as in the Sergi et al. study [16]. From this version we aimed to select empirically the best eight items for each facet balanced for reversed items. The protocol used included other measures that we did not consider in this contribution.

**Data analysis.** After descriptive analyses to investigate missing values, we, first, identified the best 8 items for each facet within each dimension (item selection), second, checked the dimensions of the whole scale (dimensional validity) and, third, established reliability of factors. All statistical analyses were performed with R 4.1.2 [32].

*Item selection.* To identify the best 8 items per facet and remove individual differences, responses were ipsatized before any recoding of the reversed items [33]. A series of Principal Component Analysis (hereafter PCA) were conducted on each of the 16 items of each facet to identify the best 5 items per facet and then to save the relative factorial scores of each factor. Then, an extension factor analysis [34, 35] was carried out to identify, among the remaining items, the additional three items for each facet: for each facet, the excluded items were correlated with the saved factorial scores of each HEXACO factor to simultaneously include items that correlated with the pertinent factor and exclude items that correlated with other factors. In both steps, the content, the direction, and the belonging of the item to the facet were considered to ensure an almost balanced number of direct and reversed items in each facet. In some cases, this procedure was reiterated to refine the analysis and reach a better outcome. Totals for dimensions and facets were calculated by summing the relative items, after recoding the reversed items by multiplying the ipsatized values by -1.

*Dimensionality.* Once the best items for each facet had been selected, two PCAs were performed first on all the items and separately on all the facets to investigate the dimensional structure of the scale. A parallel analysis in conjunction with the scree plot and the eigenvalues were used to determine the number of factors and an oblimin rotation was performed to interpret the factorial solution. Once dimensionality was established, we obtained the interim version of the scale of Study 1.

*Reliability.* Reliability of the traits of the interim version of the HEXACO-MSI-E was examined using Cronbach's alpha for each dimension and for each facet.

## Results

**Item selection.** Using the procedure described in the method, we selected from the 384-item of the HEXACO-MSI-E, the best 8 items for each facet with the only exception for the Unconventionality facet of the Openness to experience (O) because, in this case, we could identify only 6 items. The ratio direct versus reversed items was 4:4 in 16 facets, 3:5 in 2, 6:2 in 2, 5:3 in 1, 2:6 in 1, 1:7 in 1, and, finally, 3:3 for Unconventionality. This resulted in the

**Table 1. Correlations among the six factors extracted from the items and the facets of the 190-item HEXACO-MSI (Study 1).**

| Factor | On items | | | | | On facets | | | | |
|---|---|---|---|---|---|---|---|---|---|---|
| | H | E | X | A | C | H | E | X | A | C |
| H | | | | | | | | | | |
| E | -.16 | | | | | .19 | | | | |
| X | -.02 | -.04 | | | | -.01 | .09 | | | |
| A | -.27 | .05 | .09 | | | .37 | .08 | -.16 | | |
| C | .19 | -.01 | -.15 | -.17 | | -.25 | -.06 | .18 | -.21 | |
| O | -.11 | .04 | .13 | .07 | -.17 | .09 | .04 | -.14 | .09 | -.16 |

selection of a total of 190 items that correspond to the 190-item version of the HEXACO-M-SI-E (see Table A in S1 File).

**Dimensionality.**   *Factor analysis on items.* The parallel analysis, the scree plot analysis and the eigenvalues analyses on the 190 items of HEXACO-MSI-E suggested extraction of six latent factors with a cumulative explained variance of 28.1%. Eigenvalues (and percentage of explained variance) of the six components were respectively 20.17 (10.6%), 10.19 (5.4%), 7.49 (3.9%), 5.92 (3.1%), 5.27 (2.8%) and 4.26 (2.2%). The oblimin rotated solution showed that all items had an adequate loading on the unique, pertinent factor, that no item cross-loaded on not pertinent factors (Table A in S1 File) and that the factors were weakly correlated (rs < |.27|; Table 1).

*Factor analysis on facets.* Parallel, scree plot and the eigenvalues analyses on the 24 facets suggested six latent factors with a cumulative explained variance of 66.1%. Eigenvalues before rotation (and percentage of explained variance after rotation) of the six components were respectively 5.95 (24.8%), 3.12 (13.0%), 2.33 (9.3%), 1.76 (7.3%), 1.60 (6.6%) and 1.22 (5.1%). The oblimin rotated solution showed that all the facets had an adequate loading on the single, pertinent factor, that no facet cross-loaded on a not pertinent factor (Table 2) and that the six factors were modestly related (rs < |.37|; Table 1). The only facet that loads on a different factor with a value higher than .30 is Inquisitiveness on C, that anyway loads on the pertinent factor (O) much more (.60).

## Reliability

Table 2 also shows the alphas for the 6 factors based on items. They were excellent for the factors and at least sufficient for all the facets (from .64 for Gentleness to .85 for Social Self-Esteem and Organization), apart from Unconventionality for which the reliability was unsatisfactory (α = .44).

## Discussion

From the initial pool of 384 items, we identified the best 190 items by item selection and then they were subjected to exploratory factor analysis to get an interim version of HEXACO-M-SI-E. Apart from O (30 items), each factor contained 32 items and, apart from Unconventionality of O (6 items), each facet contained 8 items. While Unconventionality had 3 reversed items, 20 facets had normal and reversed items balance, 2 facets had 2 items reversed and 1 facet had 1 item reversed. The results of Study 1 were encouraging–the scale had dimensional validity and the traits were reliable and almost independent–but we planned a second study to set up and validate a definitive scale in which the number of items was the same for each facet and the ratio between direct and reversed items was balanced. Given the problems shown particularly by Unconventionality, we paid particular attention to developing new items for that facet.

**Table 2. Factor loadings (oblimin rotation) of the PCA on the facets of the 190-item HEXACO-MSI, normal and reversed items, and alpha coefficients for factors and facets based on items (Study 1).**

| Factor / Facet | H | E | X | A | C | O | Item (# reversed) | alpha |
|---|---|---|---|---|---|---|---|---|
| Honesty-Humility | | | | | | | 32 (16) | .88 |
| Sincerity | **.78** | .03 | -.10 | .00 | -.13 | -.05 | 8 (4) | .68 |
| Fairness | **.69** | .09 | -.21 | -.06 | -.21 | -.05 | 8 (4) | .78 |
| Greed Avoidance | **.77** | -.09 | .13 | .13 | .12 | .06 | 8 (4) | .75 |
| Modesty | **.78** | .04 | .13 | .11 | .15 | .00 | 8 (4) | .71 |
| Emotionality | | | | | | | 32 (16) | .85 |
| Fearfulness | .09 | **.66** | .04 | -.09 | -.22 | -.31 | 8 (4) | .76 |
| Anxiety | .05 | **.72** | .18 | -.13 | -.13 | .13 | 8 (4) | .66 |
| Dependence | -.07 | **.77** | .01 | .14 | .15 | -.07 | 8 (4) | .76 |
| Sentimentality | .05 | **.68** | -.11 | .09 | .13 | .28 | 8 (4) | .73 |
| Extraversion | | | | | | | 32 (18) | .91 |
| Social Self-Esteem | -.00 | -.18 | **-.77** | .02 | -.15 | -.12 | 8 (4) | .85 |
| Social Boldness | -.15 | -.10 | **-.74** | -.07 | .06 | .25 | 8 (4) | .71 |
| Sociability | .04 | .20 | **-.84** | .09 | .08 | -.06 | 8 (6) | .75 |
| Liveliness | .08 | -.06 | **-.79** | .10 | -.07 | -.02 | 8 (4) | .79 |
| Agreeableness | | | | | | | 32 (16) | .90 |
| Forgivingness | .04 | .13 | -.08 | **.75** | .15 | .05 | 8 (4) | .81 |
| Gentleness | .28 | -.04 | -.08 | **.61** | -.07 | .05 | 8 (4) | .64 |
| Flexibility | .08 | .01 | .02 | **.77** | -.10 | -.02 | 8 (4) | .69 |
| Patience | -.04 | -.10 | -.04 | **.81** | -.17 | -.01 | 8 (4) | .83 |
| Conscientiousness | | | | | | | 32 (12) | .92 |
| Organization | -.13 | -.01 | .03 | .19 | **-.77** | -.10 | 8 (2) | .85 |
| Diligence | .18 | -.03 | -.17 | -.08 | **-.67** | .24 | 8 (4) | .82 |
| Perfectionism | .16 | .03 | -.11 | -.13 | **-.77** | .20 | 8 (4) | .84 |
| Prudence | .08 | .02 | -.02 | .19 | **-.72** | .01 | 8 (2) | .76 |
| Openness | | | | | | | 30 (15) | .88 |
| Aesthetic Appr. | .11 | .05 | .14 | .13 | -.29 | **.65** | 8 (4) | .83 |
| Inquisitiveness | .02 | .02 | .08 | .15 | **-.37** | **.60** | 8 (4) | .78 |
| Creativity | -.11 | .15 | -.20 | -.00 | -.11 | **.69** | 8 (4) | .75 |
| Unconventionality | .03 | -.09 | .00 | -.06 | .19 | **.80** | 6 (3) | .44 |

Note. Factor loadings > .30 are in boldface;

## Study 2

As we have seen in the discussion of Study 1, the 190-item interim HEXACO-MSI-E, despite having adequate levels of dimensional validity and reliability, failed in identifying 8 items for Unconventionality facet of O and in balancing normal and reversed items within each facet. Therefore, we added items to the scale and checked again the dimensionality and reliability to reach a final version of the scale.

Study 2 had as its overall purpose to develop the final version of the scale, the HEXACO-MSI extended, and establish its psychometric properties. Specifically, the aims were to select the best eight items for each facet, thus the best thirty-two items for each factor, investigate the dimensional validity of the scale, establish its measurement invariance across males and females and across the three classes, and, finally, establish its reliability as internal consistency.

## Methods

**Sample and procedure.** The sample was composed by 1175 middle school students (52.3% Females, Mean age = 12.03, SD = 0.89, age range = 10–14), 34.1% attending the 6[th], 33.9% the 7[th] and 32% the 8[th] grade. All the procedures were the same as those used in Study 1 apart from differences in content as reported below.

**Measures.** The 219 items of the HEXACO-MSI-E. After the background information, we administered a version of the HEXACO-MSI-E consisting of 219 items. First, we added 29 items to the 190-item interim scale coming from the Study 1 to fix the problems mentioned above. Particularly, we added items to the facets Fearfulness (3) and Sentimentality (2) of E, Social Boldness (3) of X, Flexibility (4) and Gentleness (3) of A, Diligence (3) and Perfectionism (3) of C as well as in Unconventionality (8) of O. The resulting 219-item scale of the HEXACO-MSI-E in Study 2 was then administrated to the sample of adolescents. Similarly to Study 1, the protocol used, along with the marks at the end of the year, included other measures that we did not use in this contribution.

**Data analysis.** After descriptive analyses to investigate missing values, an exploratory factor analysis was conducted on each facet in which we added items, to identify the best 8 items for each facet (item selection). Then, PCA on items and Confirmatory Factor Analysis (CFA) on facets were executed to establish dimensional validity. Finally, the Measurement Invariance analysis and the reliability analysis were performed respectively to test the robustness of the latent structure and its reliability.

*Item selection.* Responses were preliminarily ipsatized as in Study 1 [33]. Then, to improve the validity of the scale and identify the best 8 items per facet, balanced for the number of reversed items, a one-factor PCA was carried out on the items of each modified facet: Fear (11 items) and Sentimentality (10 items) of E, Social Boldness of X (11 items), Flexibility (12 items) and Gentleness (11 items) of A, Diligence (11 items) and Perfectionism (11 items) of C, and Unconventionality of O (14 items). Item selection was guided by the direction of association with the component, loading strength and item content.

*Dimensional validity.* Once obtained the best 8 items for each facet, a PCA was performed on the ipsatized items to explore the dimensionality of the 192-item scale, whereas a CFA was carried out on the facets computed on the ipsatized items to verify the 6-factor latent structure. The facets scores were computed as the sum of the ipsatized scores of the individual items. In the PCA, the latent solution was determined by means of parallel analysis in conjunction with the scree plot and the eigenvalues analyses, whereas the oblimin rotation was performed to interpret the factorial solution. In the CFA a six-correlated latent-factors model was fitted. To test the goodness-of-fit of the models [36], the maximum likelihood (ML$X^2$) test-statistics, the root mean square error of approximation index (RMSEA), and the comparative fit index (CFI) were used. Finally, the modification indices of the final model were analyzed in order to recognize any additional parameters to be estimated.

*Measurement invariance (MI).* To evaluate the robustness of the latent structure of the 192-item HEXACO-MSI-E, the MI was tested across: (a) Sex; and (b) Classes. The following hierarchically structured multi-group tests were carried out to establish the specific invariance of the measure: (a) configural invariance; (b) metric or weak factorial invariance; (c) scalar or strong factorial invariance; and (d) residual or strict or invariant uniqueness. Means and covariance matrices were considered to test MI (MACS) [37].

*Reliability.* Reliability of each dimension and facet was evaluated with both Cronbach's alpha and omega (ωt) [38].

Confirmatory factor analyses (CFAs) and measurement invariance (MI) analysis were performed using Lisrel 8.71 [39]. All the other analyses were performed using R 4.1.2 [32].

## Results

**Item selection of the modified facets.** Eight unidimensional PCA on the items coming from both the 190-item version of Study 1 and on the ones generated for the Study 2 were executed on the 8 facets mentioned in the item selection in the Method section. It gave as a result an equal number of normal and reversed items within each facet (4:4). With respect to the 190-item version of Study 1, the items of H-H were exactly the same, 4 items of E were substituted (2 from Fearfulness and 2 from Sentimentality), 3 of X (all from Social Boldness), 5 from A (3 from Flexibility and 2 from Gentleness), 4 from C (2 from Diligence and 2 from Perfectionism); finally, the items of O were confirmed apart for the items of Unconventionality: 3 of them were substituted and 2 were added in order to have a total of 8 items for that facet. In total, 19 items were changed and 2 more were added. At this stage, the scale was formed by 192 items (8 per facet and 32 per factor).

**Dimensional validity.** *Exploratory factor analysis on items.* Scree plot and eigenvalues as well as parallel analyses on the 192 items of HEXACO-MSI-E suggested extraction of six latent factors with a cumulative explained variance of 33.0%. Eigenvalues (before rotation) and percentage of explained variance (after rotation) of the first six components were 22.5 (6.3%), 12.29 (5.4%), 9.22 (5.3%), 6.49 (4.6%), 4.91 (4.8%), and 4.45 (4.7%). (Table B in S1 File) shows the factor loadings of the oblimin 6-factor rotated solution. As expected, the vast majority of items had an adequate loading on the relative dimension and very low on not pertinent dimensions, with minor exceptions. Table 3 shows that the six factors were weakly related (rs < |.28|).

*Confirmative factor analysis on facets.* CFA on facets shows that the 6-factor model had sufficient fit indices. The analysis of the modification indices revealed the significance of some additional parameters. Accordingly, three cross-loadings and five covariances between errors were considered, RMSEA = .072; 90% CI [.069, .076]; CFI = .940, $MLX^2$(229, N = 1,175) = 1691.85, p < .001. In the final model the facets that loaded on a different factor with a value higher than .30 were Sentimentality of E on X, Sociability of X on E, and Gentleness of A on H. All the facets presenting a significant loading on a secondary dimension showed that the loading on the pertinent factor was higher. As regards the error covariances, the following correlated error terms were considered: between Greed Avoidance of H and Modesty of H, Social Boldness of X and Fear of E, Social Boldness of X and X-Liveliness of X, Flexibility of A and Patience of A and between Unconventionality of O and Creativity of O. The indicators and their standardized factor loadings are reported in Table 4.

As shown in Table C in S1 File, latent factors were associated. Correlation ranged from |.097| to |.665| with a mean of .322. Relevant correlations (>|.40|) were observed between H and A, H and C, E and X, and C and O.

**Table 3. Correlations among the six factors extracted from the items (EFA) of the 192-item HEXACO-MSI-E (Study 2).**

| Factor | H | E | X | A | C |
|--------|------|------|------|------|------|
| H | | | | | |
| E | -.19 | | | | |
| X | -.06 | -.09 | | | |
| A | -.27 | .07 | .18 | | |
| C | .22 | -.02 | -.18 | -.17 | |
| O | -.15 | .04 | .13 | .05 | -.28 |

Note. H = Honesty/Humility; E = Emotionality; X = Extraversion; A = Agreeableness; C = Conscientiousness; O = Openness to Experience.

**Table 4. Factor loadings of the 6-factor CFA on the facets of the 192-item HEXACO-MSI-E (Study 2).**

| Factor / Facet | H | E | X | A | C | O |
|---|---|---|---|---|---|---|
| Honesty-Humility | .76 | | | | | |
| Sincerity | | | | | | |
| Fairness | .79 | | | | | |
| Greed Avoidance | .61 | | | | | |
| Modesty | .65 | | | | | |
| Emotionality | | .53 | | | | |
| Fearfulness | | | | | | |
| Anxiety | | .69 | | | | |
| Dependence | | .69 | | | | |
| Sentimentality | | .78 | .30 | | | |
| Extraversion | | | .80 | | | |
| Social Self-Esteem | | | | | | |
| Social Boldness | | | .79 | | | |
| Sociability | | .37 | .85 | | | |
| Liveliness | | | .87 | | | |
| Agreeableness | | | | .70 | | |
| Forgivingness | | | | | | |
| Gentleness | .38 | | | .54 | | |
| Flexibility | | | | .74 | | |
| Patience | | | | .76 | | |
| Conscientiousness | | | | | .52 | |
| Organization | | | | | | |
| Diligence | | | | | .86 | |
| Perfectionism | | | | | .89 | |
| Prudence | | | | | .65 | |
| Openness to Experience | | | | | | .82 |
| Aesthetic Appreciation | | | | | | |
| Inquisitiveness | | | | | | .77 |
| Creativity | | | | | | .53 |
| Unconventionality | | | | | | .28 |

Note. H = Honesty/Humility; E = Emotionality; X = Extraversion; A = Agreeableness; C = Conscientiousness; O = Openness to Experience.

*Measurement invariance (MI).* The MI of the HEXACO-MSI-E was tested across: (a) sex; and (b) classes (Table 5). In the MI analysis, the latent structure used in the previous CFA analyses was used. The results showed a full metric invariance across the three considered factors.

**Reliability.** All dimensions of the HEXACO-MSI-E had at least good levels of internal consistency, as shown by Omegas and alpha indexes for internal consistency in Table 6. As for the facets, 13 indexes range between .80 and .90, 7 between .70 and .79 and 4 between .60-.69.

**Descriptive statistics of the HEXACO-MSI-E.** Table 7 shows descriptive statistics based on raw scores of the items of the final version of the HEXACO-MSI-E for the whole sample and for both males and females (the intercorrelations between scale scores of the dimensions of the HEXACO-MSI-E are reported in Table D in S1 File).

## Discussion

The results of Study 2 indicated that we fulfilled the general aim of developing a final version of the HEXACO-MSI-E with good psychometric properties, able to improve over the interim

**Table 5. Invariance analysis as a function of sex (Males vs females), and class (C1 "I media" vs C2 "II media" vs C3 "III media") of the HEXACO-MSI-E: Multi-group hierarchical confirmatory factor analyses goodness-of-fit indices.**

| Model | RMSEA | CFI | NNFI | $ML\chi^2$ | df | $ML\chi^2_{diff}$ | $df_{diff}$ | $CFI_{diff}$ | $RMSEA_{diff}$ |
|---|---|---|---|---|---|---|---|---|---|
| | | | | Sex (a) | | | | | |
| Model A | .078 | .936 | .924 | 2446.78*** | 464 | – | – | – | – |
| Model B | .077 | .935 | .926 | 2473.10*** | 482 | 26.32 | 18[a] | .001 | -.001 |
| Model C | .083 | .922 | .914 | 2773.64*** | 500 | 300.54*** | 18[b] | .013 | .006 |
| Model D | .083 | .917 | .913 | 2929.73*** | 529 | 156.09*** | 29[c] | .005 | 0 |
| | | | | Class (b) | | | | | |
| Model A | .076 | .939 | .927 | 1766.55*** | 464 | – | – | – | – |
| Model B | .075 | .938 | .929 | 1788.05*** | 482 | 27.50 | 18[a] | .001 | -.001 |
| Model C | .073 | .938 | .932 | 1802.61*** | 500 | 14.56 | 18[b] | 0 | -.002 |
| Model D | .071 | .939 | .937 | 1815.32*** | 529 | 12.71 | 29[c] | -.001 | -.002 |

Note. Sex: Males n = 560, Females n = 615. Class: C1 n = 402; C2 n = 400; C3 n = 373. Model A: six-factor configural invariance (CI). Model B: six-factor CI and metric invariance (MI). Model C: six-factor CI, MI, and scalar invariance (SI). Model D: six-factor CI, MI, SI, and invariant uniqueness (IU). [a]The reference model is Model A. [b]The reference model is Model B. [c]The reference model is Model C.

**p < .01;

***p < .001.

190-item scale developed in Study 1. The final version of the scale, the HEXACO-MSI-E, is therefore formed by 192 items, balanced for direct and reversed items within each facet and, consequently, for each factor. The dimensionality of the scale was established by an exploratory factor analysis on the items, that produced a weakly correlated six-factor solution, and confirmed by a confirmative factor analysis on the facets. In the latter solution, the correlations between factors were higher than those observed in the exploratory solution, most likely because they were extracted from the facets and due to the constraints of the model. In fact, the correlations between factors in unconstrained PCA at the level of items were weak and in line with the results of Study 1. While we think that the most important reason for the correlations among factors of the CFA is the constraints imposed by the model, they might also be partially due to a strong impact of individual differences in responding in a desirable versus undesirable way in adolescents. The six-dimensional structure was observed to be invariant across males and female, and across the three classes. Also, the reliability as internal consistency was very good for the factors and at least satisfying for the facets. Higher correlation than expected resulted in the correlations of the dimensions coming from the CPA on facets.

## Study on stability of traits across Study 1 and 2

One final aim of this contribution was establishing test-retest reliability with one year as the interval between the tests to check the stability of the six traits in adolescents.

### Methods

**Test-retest procedures and temporal stability of traits.** We wanted to establish a test-retest study with two collections of the same participants to Study 1 and 2 ($t_1$ and $t_2$, respectively) with an interval of one year (April-May 2020 and 2021). Indeed, during the second recruitment, we purposely asked two schools collaborating on the first study to renew their collaboration for the second study. This provided the opportunity to have the same 182 adolescents about one year after the first data collection (53.3% females, $t_1$: Mean age = 11.70,

**Table 6. Reliability coefficients of the 192-item HEXACO-MSI-E (Study 2) and stability indexes with 95% CI for the test-retest study (Study 1 and 2).**

| Factor / Facet | # Item Total (Reversed) | Study 2 omega | Study 2 alpha | Test-Retest Absolute ICC | Test-Retest 95% CI |
|---|---|---|---|---|---|
| H (Honesty-Humility) | 32 (16) | .91 | .90 | .67 | [.58; .74] |
| Sincerity | 8 (4) | .76 | .69 | .54 | [.42; .63] |
| Fairness | 8 (4) | .85 | .82 | .50 | [.38; .60] |
| Greed Avoidance | 8 (4) | .82 | .76 | .64 | [.55; .72] |
| Modesty | 8 (4) | .82 | .74 | .58 | [.47; .67] |
| E (Emotionality) | 32 (16) | .90 | .89 | .65 | [.56; .73] |
| Fearfulness | 8 (4) | .82 | .78 | .64 | [.55; .72] |
| Anxiety | 8 (4) | .75 | .67 | .51 | [.39; .61] |
| Dependence | 8 (4) | .88 | .82 | .62 | [.53; .71] |
| Sentimentality | 8 (4) | .85 | .81 | .56 | [.45; .65] |
| X (Extraversion) | 32 (16) | .94 | .93 | .66 | [.52; .75] |
| Social Self-Esteem | 8 (4) | .90 | .87 | .62 | [.49; .72] |
| Social Boldness | 8 (4) | .85 | .80 | .56 | [.44; .65] |
| Sociability | 8 (4) | .87 | .82 | .63 | [.54; .72] |
| Liveliness | 8 (4) | .87 | .83 | .57 | [.33; .71] |
| A (Agreeableness) | 32 (16) | .93 | .92 | .72 | [.64; .79] |
| Forgivingness | 8 (4) | .88 | .84 | .57 | [.46; .66] |
| Gentleness | 8 (4) | .83 | .78 | .54 | [.33; .69] |
| Flexibility | 8 (4) | .77 | .69 | .51 | [.38; .62] |
| Patience | 8 (4) | .88 | .86 | .64 | [.55; .72] |
| C (Conscientiousness) | 32 (16) | .94 | .93 | .76 | [.69; .81] |
| Organization | 8 (4) | .92 | .88 | .71 | [.63; .78] |
| Diligence | 8 (4) | .85 | .82 | .70 | [.61; .77] |
| Perfectionism | 8 (4) | .89 | .86 | .69 | [.59; .76] |
| Prudence | 8 (4) | .85 | .78 | .59 | [.49; .68] |
| O (Openness to Experience) | 32 (16) | .91 | .89 | .68 | [.60; .75] |
| Aesthetic Appreciation | 8 (4) | .89 | .85 | .65 | [.55; .72] |
| Inquisitiveness | 8 (4) | .85 | .81 | .64 | [.54; .72] |
| Creativity | 8 (4) | .84 | .77 | .63 | [.53; .71] |
| Unconventionality | 8 (4) | .75 | .61 | .33 | [.19; .45] |

SD = 0.70; $t_2$: Mean age = 12.53, SD = 0.65): 53.8% of them was attending the 6th grade and 46.2% the 7th grade during the first data collection.

For the $t_1$ measurement, we used the 190-item interim version of the Study 1, for $t_2$, the 192-item final version of the Study 2. Test-retest reliability was established by absolute estimates of Intraclass Correlation Coefficient (ICC) and confidence interval based on a single rating (k = 2) by a 2-way mixed effects model [40] for both factors and facets. SPSS was used for calculating ICCs.

## Results

**Stability.** The absolute ICCs for each trait of the HEXACO-MSI-E between $t_1$ and $t_2$ ranged from .65 to .76 (last two columns of Table 6) suggesting that HEXACO scores of traits of children were remarkably stable after one year and considering that participants were adolescents in a full developmental period of life. Out of all the ICCs for the facets, 2 range .70.-79, 10 range .60-.69, 11 range .50-.59 and only one results low, i.e., .33 (Unconventionality of O).

**Table 7. Descriptive statistics (mean and SD) for the entire sample (N = 1175), for males (N = 560) and females (N = 615).**

| Factor / Facet | Total sample (N = 1175) | | Male (N = 560) | | Female (N = 615) | |
|---|---|---|---|---|---|---|
| | Mean | SD | Mean | SD | Mean | SD |
| H (Honesty-Humility) | 3.73 | .66 | 3.73 | .63 | 3.72 | .69 |
| Sincerity | 3.76 | .77 | 3.79 | .75 | 3.73 | .79 |
| Fairness | 4.13 | .85 | 4.11 | .85 | 4.14 | .84 |
| Greed Avoidance | 3.17 | .86 | 3.17 | .82 | 3.18 | .89 |
| Modesty | 3.84 | .79 | 3.85 | .73 | 3.84 | .83 |
| E (Emotionality) | 3.24 | .66 | 3.10 | .61 | 3.38 | .67 |
| Fearfulness | 2.95 | .90 | 2.83 | .85 | 3.06 | .94 |
| Anxiety | 3.57 | .72 | 3.43 | .67 | 3.70 | .74 |
| Dependence | 3.06 | .91 | 2.98 | .85 | 3.14 | .96 |
| Sentimentality | 3.39 | .96 | 3.15 | .88 | 3.62 | .97 |
| X (Extraversion) | 3.52 | .76 | 3.69 | .68 | 3.37 | .80 |
| Social Self-Esteem | 3.42 | .97 | 3.61 | .89 | 3.24 | 1.00 |
| Social Boldness | 3.23 | .92 | 3.35 | .86 | 3.11 | .95 |
| Sociability | 3.89 | .86 | 4.09 | .73 | 3.71 | .92 |
| Liveliness | 3.55 | .89 | 3.70 | .82 | 3.41 | .93 |
| A (Agreeableness) | 3.17 | .73 | 3.38 | .65 | 2.98 | .75 |
| Forgivingness | 3.07 | .94 | 3.35 | .86 | 2.81 | .93 |
| Gentleness | 3.68 | .82 | 3.77 | .74 | 3.60 | .87 |
| Flexibility | 3.13 | .75 | 3.34 | .67 | 2.94 | .78 |
| Patience | 2.79 | 1.04 | 3.06 | .99 | 2.56 | 1.02 |
| C (Conscientiousness) | 3.33 | .78 | 3.29 | .77 | 3.37 | .78 |
| Organization | 3.14 | 1.08 | 3.11 | 1.06 | 3.16 | 1.09 |
| Diligence | 3.59 | .87 | 3.52 | .86 | 3.65 | .87 |
| Perfectionism | 3.34 | .99 | 3.23 | .99 | 3.45 | .98 |
| Prudence | 3.25 | .88 | 3.29 | .84 | 3.21 | .92 |
| O (Openness to Experience) | 3.31 | .66 | 3.25 | .62 | 3.36 | .69 |
| Aesthetic Appreciation | 3.10 | 1.01 | 3.00 | 1.00 | 3.20 | 1.00 |
| Inquisitiveness | 3.32 | .95 | 3.40 | .93 | 3.24 | .96 |
| Creativity | 3.51 | .85 | 3.42 | .80 | 3.59 | .88 |
| Unconventionality | 3.31 | .68 | 3.18 | .62 | 3.42 | .72 |

Paired t-test comparisons of the six dimensions at $t_1$ and $t_2$ are shown in Table E in S1 File. Only one results significant with FDR correction: X decreases after one year (for the analogue comparisons for facets see Table F in S1 File).

## Discussion

Considering the substantial time intercurrent between the first and the second data collection and considering that the two scales are made up of some different items, it can be concluded that the traits and facets are substantially stable after one year. The unique exception is Unconventionality of O which is the only facet with low stability.

## General discussions and conclusion

This contribution is focused on developing and proposing a novel instrument for evaluating adolescent personality with desirable psychometric features. To our knowledge, there is not a so comprehensive scale for that purpose. Starting from a very large and representative pool of

items, which included all the items from which the older HEXACO-MSI was developed, the IPIP [29] and the HEXACO-SPI [30], we established a 384-item pool with the same number of items within each facet and each dimension and with balanced reversed items. From this pool, we developed, throughout two studies, a novel 192-item scale, with increased number of items (from the 48 of the older version of Sergi et al. to the 192 of the extended), same number of items within each facet (8) and, thus, each dimension (32), representativeness of the items with respect to facets and dimensions, and, finally, perfect equilibrium between positive and negative items within facets and dimensions. In the following, we will briefly list the positive outcomes of the novel HEXACO-Medium School Inventory Extended (HEXACO-MSI-E) and then discuss issues that could be addressed in future works.

The results show a six-dimensional structure of personality coherent with the HEXACO model in the samples of adolescents of Study 1 and 2 (N = 714 and N = 1175) both with exploratory and confirmative factor analysis, applied to both items and facets. Each trait and each facet, in both studies, show substantial internal consistency (reliability). The identified six-factor structure remains equivalent across three random chosen groups of adolescents, males and females and, finally, the three classes of the middle school that composed the sample of the second study. This pattern of results provides strong evidence for the coherence of the structure, also with different classes of age during adolescence, and for the generalization of the factor structure to the target population. Finally, the interval between the two surveys to establish stability was 1 year and the results showed that traits and facets were substantially stable when passing from the 6th to the 7th grade. In conclusion, our study shows that basic elements of personality are present and maintained during middle school with both cross-sectional (see above the invariance across age classes) and longitudinal data (test-retest). The results also show the overall good psychometric qualities of the HEXACO-MSI-E. Notwithstanding we used hypsatized scores for scale validation purposes, the scale scores can simply be calculated as averages of the raw item scores.

One interesting unexpected result of this study regards the facet of Unconventionality of O. In Study 1, we were able to identify only 6 items (against the 8 of each other facet), with relatively low factor loadings on O (all between -0.24 and 0.42), although the loading of the facet on O was satisfying. We also did find low internal consistency (.44). Therefore, we took great care of improving on the measure of this facet in Study 2 and we added 8 more items for a total of 14. Nevertheless, the factor loadings on O of the best 8 items selected by the analysis were not very high (between -.38 and .49) and the factor loading of the facet on O was low (.22). It was the only facet whose loading was lower than .30. Although the internal consistency in Study 2, measured with alpha and omega, was sufficient, it still was one of the lowest. Finally, test-retest on the facet shows that the stability of Unconventionality was the lowest of all the facets. We can speculate on the possible reasons why this pattern of results, considering any case that reliability of Unconventionality in adults seems to have similar values (e.g., in two recent studies, it was .52 and .54; [41, 42]). One hypothesis could be that Unconventionality in the age range we have studied is a not yet structured aspect of the personality. Low consistency within the measure, low stability and low adherence to O could reflect just an age phase in which the aspects of unconventionality are going to develop but they are not yet completely maturated. Another possibility is that unconventionality in adolescence could be something different from its corresponding dimension in adults, particularly, it could be expressed in different ways or it could as well express conformism even if toward a subgroup. Third, we should any case recognize that item content of Unconventionality of the HEXACO-MSI-E is partially different from the original adult scale that emphasizes more eccentricity, nonconformity and appreciation of oddness. Future studies should contemplate these considerations for both better understanding the role of Unconventionality in adolescence and trying

to improve this facet as well as replicate the HEXACO-MSI-E in different samples of adolescents.

## Supporting information

**S1 File. Supplementary materials.**
(DOCX)

## Author Contributions

**Conceptualization:** Augusto Gnisci, Marco Perugini, Vincenzo Paolo Senese.

**Data curation:** Augusto Gnisci, Francesca Mottola, Vincenzo Paolo Senese, Ida Sergi.

**Formal analysis:** Augusto Gnisci, Marco Perugini, Vincenzo Paolo Senese.

**Funding acquisition:** Vincenzo Paolo Senese, Ida Sergi.

**Investigation:** Francesca Mottola, Marco Perugini.

**Methodology:** Augusto Gnisci, Francesca Mottola, Marco Perugini, Vincenzo Paolo Senese.

**Project administration:** Vincenzo Paolo Senese, Ida Sergi.

**Resources:** Vincenzo Paolo Senese, Ida Sergi.

**Software:** Ida Sergi.

**Supervision:** Augusto Gnisci, Marco Perugini.

**Validation:** Augusto Gnisci, Francesca Mottola, Marco Perugini, Vincenzo Paolo Senese.

**Writing – original draft:** Augusto Gnisci, Francesca Mottola, Vincenzo Paolo Senese, Ida Sergi.

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
