## [Decision Letter · Decision Letter 0]

1 Dec 2022

PONE-D-22-29099Development and validation of an instrument to measure personality in adolescence: The HEXACO Medium School Inventory Extended (MSI-E)PLOS ONE

Dear Dr. Gnisci,

Thank you for submitting your manuscript to PLOS ONE. After careful consideration, we feel that it has merit and we invite you to consider points raised during the review process.

We look forward to receiving your revised manuscript.

Kind regards,

Frantisek Sudzina

Academic Editor

PLOS ONE

Journal Requirements:

Reviewers' comments:

Reviewer's Responses to Questions

**Comments to the Author**

1. Is the manuscript technically sound, and do the data support the conclusions?

Reviewer #1: Yes

Reviewer #2: Yes

2. Has the statistical analysis been performed appropriately and rigorously? 

Reviewer #1: Yes

Reviewer #2: Yes

3. Have the authors made all data underlying the findings in their manuscript fully available?

Reviewer #1: Yes

Reviewer #2: Yes

4. Is the manuscript presented in an intelligible fashion and written in standard English?

Reviewer #1: Yes

Reviewer #2: Yes

5. Review Comments to the Author

Reviewer #1: Review of PONE-D-22-29099, “Development and validation of an instrument to measure personality in adolescence: The HEXACO Medium School Inventory Extended (MSI-E)”

As its title suggests, this manuscript reports on the development and validation of a personality inventory for middle-school-aged adolescents.

In my opinion, the procedures and analyses reported here, as well as the background to and the discussion of those elements, have all been undertaken very competently. It’s clear that the authors have a deep understanding of psychological test construction, of statistical and psychometric methods, and of personality psychology in general (as well as the HEXACO model in particular). Furthermore, the instrument itself appears to show very good psychometric properties, and it’s of course very useful to have a full-length HEXACO instrument suitable for adolescents. For these reasons, I think that the manuscript is highly deserving of publication.

Below, I list various suggestions for relatively minor revisions to the manuscript:

1. On page 2, lines 50-51, the last part of the sentence seems to repeat the defining content of Honesty-Humility, which is already given in the previous sentence; therefore, this second part of the latter sentence could be deleted. Also, in the first part of this latter sentence, it would be good to acknowledge that Honesty-Humility often shows modest correlations with FFM Agreeableness.

2. On page 6, the phrase “in two close sessions” could be clarified, perhaps to say “in two sessions about XXX days apart”

3. On page 7, where ipsatization of item responses is mentioned, it should be specified as to whether this was done before any recoding of reversed items. (It should be done before such recoding.)

4. On page 8, in describing the ratio of non-reversed to reversed items in each facet, the segment “3:3 for O” should be “3:3 for Unconventionality”, because this is referring to this particular facet rather than to the entire factor scale.

5. On page 9, the percentages of variance given for the dimensions seem to be for the original eigenvalues and not for the sizes of the rotated dimensions (in contrast, the percentages on page 13 do seem to be for the rotated dimensions)

6. On page 11, line 238, “CPA on items” should be “PCA on items”, and on page 18, line 350, “CCIs” should be “ICCs”.

7. The results for the stability of scores across the one-year period are interesting. It might also be interesting to note briefly any facet or factor scales for which some non-trivial mean-level changes were observed.

8. It’s also interesting that the CFA-based factor intercorrelations tend to be rather high. One wonders whether this might reflect stronger individual differences, in adolescent samples, in a tendency to respond desirably versus undesirably.

9. Regarding the Unconventionality scale, I tend to think that the item content of this scale departs somewhat from the emphasis on eccentricity, nonconformity, and appreciation of oddness that is seen in the original HEXACO facet scale. Perhaps a minor future project—but not at all an important priority, and certainly not needed for the present report—would be to try to revise this scale along these lines.

10. It could be of interest to include a supplementary table showing the means, SDs, reliabilities, and intercorrelations of the factor- and facet-level scales. Perhaps this table could be repeated so that separate results for boys and girls are also reported. But I would leave this to the authors’ judgment.

Reviewer #2: The authors report on psychometric properties of the HEXACO-MIS Extended (HEXACO-MSI-E), which was revised from its earlier version (Sergi et al., 2019). Based on the two large samples of Italian adolescents (age ranging from 10 to 14), the authors showed that the HEXACO-MSI-E is internally consistent, factorially valid, and temporally stable. In addition, by developing 24 facet scales subsumed within the 6 factors, the HEXACO-MSI-E now provides a fine-grained description of adolescent personality traits. Given the importance of examining personality traits of this age group (e.g., school yard bullying), the development of the HEXACO-MIS-E would be extremely useful for personality researchers.

Although the present report is fairly comprehensive already, I have several suggestions, which the authors may or may not consider in revising the current manuscript.

1. Given this is the first report on psychometric properties of the inventory, it would be very useful to provide basic descriptive statistics of the scale scores (i.e., means and SDs). If the authors choose to do this,

1-1. They may want to specify how to compute the scale scores (average across raw scores or ipsatized scores)

1-2. They may also provide correlations between the six scale scores (as opposed to the six factor scores). This could be useful considering that many subsequent studies are likely to use scale scores (rather than factor scores).

1-3. Means and SDs can be reported separately for men and women. It would be interesting to see if the size of sex differences in scale scores is different from those obtained from adult samples.

In abstract, the authors said “test-retest reliability”, but given that the time interval involved in this study is fairly long, I am wondering if the authors should use the term “temporal stability” (throughout the manuscript).

Line 50: “largely unrelated to” may be somewhat strong. It might be more accurate to state “only modestly related to”.

Line 114, wouldn’t it be better to spell out “Mage” (i.e., “Mean age”)?

Line 165, it was unclear whether facet scale scores were computed based on raw scores or ipsatized scores. Given that the scales are well balanced in direction, I am wondering if scales scores can simply be computed using raw scores. In any case, the authors may want to make it clear for future users of the inventory whether they recommend using ipsaized or raw scores in computing the scale scores.

Line 204, the Coefficient alpha for the Unconventionality facet scale have been found to be fairly low from adult samples as well (typically in the .50s).

I don’t see the need to conduct measurement invariance analyses across three randomly selected subgroups and these analyses could be removed for the sake of simplicity.

I think the term, “exploratory” factor analysis is more commonly used than is “explorative” factor analysis.

6. PLOS authors have the option to publish the peer review history of their article (what does this mean?). If published, this will include your full peer review and any attached files.

Reviewer #1: No

Reviewer #2: No

---

## [Author Response · Author response to Decision Letter 0]

27 Dec 2022

PONE-D-22-29099

Development and validation of an instrument to measure personality in adolescence: The HEXACO Medium School Inventory Extended (MSI-E)

PLOS ONE

Reviewer #1

Reviewer #1: Review of PONE-D-22-29099, “Development and validation of an instrument to measure personality in adolescence: The HEXACO Medium School Inventory Extended (MSI-E)”

As its title suggests, this manuscript reports on the development and validation of a personality inventory for middle-school-aged adolescents.

In my opinion, the procedures and analyses reported here, as well as the background to and the discussion of those elements, have all been undertaken very competently. It’s clear that the authors have a deep understanding of psychological test construction, of statistical and psychometric methods, and of personality psychology in general (as well as the HEXACO model in particular). Furthermore, the instrument itself appears to show very good psychometric properties, and it’s of course very useful to have a full-length HEXACO instrument suitable for adolescents. For these reasons, I think that the manuscript is highly deserving of publication.

A: We thank R1 for their words and for the provided suggestions that improved the paper very much. As detailed below, we have followed almost all the suggestions by R1.

Below, I list various suggestions for relatively minor revisions to the manuscript:

1. On page 2, lines 50-51, the last part of the sentence seems to repeat the defining content of Honesty-Humility, which is already given in the previous sentence; therefore, this second part of the latter sentence could be deleted. Also, in the first part of this latter sentence, it would be good to acknowledge that Honesty-Humility often shows modest correlations with FFM Agreeableness.

A: Done. Thanks for the suggestion.

2. On page 6, the phrase “in two close sessions” could be clarified, perhaps to say “in two sessions about XXX days apart”

A: We have specified the average days (and SD) in which the adolescents completed the questionnaire (1.4, DS=.69) with 97.8% within 2 days.

3. On page 7, where ipsatization of item responses is mentioned, it should be specified as to whether this was done before any recoding of reversed items. (It should be done before such recoding.)

A: Yes, the ipsatization was done before the recoding of the reversed items. We have specified it in the text and added the way we calculated the totals of the dimensions and facets (i.e., multiplying the ipsatized value by -1).

4. On page 8, in describing the ratio of non-reversed to reversed items in each facet, the segment “3:3 for O” should be “3:3 for Unconventionality”, because this is referring to this particular facet rather than to the entire factor scale.

A: Done. Thanks.

P 5. On page 9, the percentages of variance given for the dimensions seem to be for the original eigenvalues and not for the sizes of the rotated dimensions (in contrast, the percentages on page 13 do seem to be for the rotated dimensions)

A: Indeed, in both Study 1 and 2, we reported the eigenvalues before rotation and the percentage of explained variance after rotation for each of the six components (as reported at p. 9 of the paper).

6. On page 11, line 238, “CPA on items” should be “PCA on items”, and on page 18, line 350, “CCIs” should be “ICCs”.

A: Done.

7. The results for the stability of scores across the one-year period are interesting. It might also be interesting to note briefly any facet or factor scales for which some non-trivial mean-level changes were observed.

A: In the SM we have added the paired t-test comparisons, corrected with FDR, between t1 and t2 for both the dimensions and the facets. In the text we refer to these tables to invite the interested reader to appreciate the differences in time in levels of personality traits (Table E and F). However, in line with what observed by R1 (“non-trivial mean-level changes”), in the text of the paper we have only described the unique significant difference as for the dimensions, that is the significant decrease of X after one year (from t1 to t2). 

It’s also interesting that the CFA-based factor intercorrelations tend to be rather high. One wonders whether this might reflect stronger individual differences, in adolescent samples, in a tendency to respond desirably versus undesirably.

A: We think that the main reason why the CFA-based correlation tends to be high is mainly due to the constraints imposed by the model given that the correlations between factors based on PCA on items are low. However, we recognize that social desirability may have a role. Therefore, when discussing the correlations, we added this text: “…they might also be partially due to a strong impact of individual differences in responding in a desirable versus undesirable way in adolescents”.

9. Regarding the Unconventionality scale, I tend to think that the item content of this scale departs somewhat from the emphasis on eccentricity, nonconformity, and appreciation of oddness that is seen in the original HEXACO facet scale. Perhaps a minor future project—but not at all an important priority, and certainly not needed for the present report—would be to try to revise this scale along these lines.

A: We thank R1 for this consideration that we have briefly incorporated in the discussion as third possible explanation for our results on Unconventionality.

10. It could be of interest to include a supplementary table showing the means, SDs, reliabilities, and intercorrelations of the factor- and facet-level scales. Perhaps this table could be repeated so that separate results for boys and girls are also reported. But I would leave this to the authors’ judgment. 

A: As correctly noted by R1 (and R2), we failed to report descriptive statistics in the first version of the paper. Now, the paper reports a novel paragraph at the end of the results of Study 2, titled “Descriptive Statistics of the HEXACO-MSI-E” that includes a new table (Table 7) with means and SD for the whole sample, the males and the females, for both dimensions and facets. The table of intercorrelations between scale scores for the dimensions is shown in a new table in the SM (Table D).

Reviewer #2

Reviewer #2: The authors report on psychometric properties of the HEXACO-MIS Extended (HEXACO-MSI-E), which was revised from its earlier version (Sergi et al., 2019). Based on the two large samples of Italian adolescents (age ranging from 10 to 14), the authors showed that the HEXACO-MSI-E is internally consistent, factorially valid, and temporally stable. In addition, by developing 24 facet scales subsumed within the 6 factors, the HEXACO-MSI-E now provides a fine-grained description of adolescent personality traits. Given the importance of examining personality traits of this age group (e.g., school yard bullying), the development of the HEXACO-MIS-E would be extremely useful for personality researchers. 

A: We thank R2 for their words and for the provided suggestions that improved the paper very much. As detailed below, we have followed almost all the suggestions by R2.

Although the present report is fairly comprehensive already, I have several suggestions, which the authors may or may not consider in revising the current manuscript.

1. Given this is the first report on psychometric properties of the inventory, it would be very useful to provide basic descriptive statistics of the scale scores (i.e., means and SDs). If the authors choose to do this,

1-1. They may want to specify how to compute the scale scores (average across raw scores or ipsatized scores)

1-2. They may also provide correlations between the six scale scores (as opposed to the six factor scores). This could be useful considering that many subsequent studies are likely to use scale scores (rather than factor scores).

1-3. Means and SDs can be reported separately for men and women. It would be interesting to see if the size of sex differences in scale scores is different from those obtained from adult samples.

A: As correctly noted by R2 (and R1), we failed to report descriptive statistics in the first version of the paper. Now, the paper reports a novel paragraph at the end of the results of Study 2, titled “Descriptive Statistics of the HEXACO-MSI-E” that includes a new table (Table 7) with means and SD for the whole sample, the males and the females, for both dimensions and facets. The table of intercorrelations between scale scores for the dimensions is shown in a new table in the SM (Table D). Moreover, we have specified in the conclusions that, notwithstanding we used hypsatized scores for scale validation purposes, the scale scores can simply be calculated as averages of the raw item scores.

2. In abstract, the authors said “test-retest reliability”, but given that the time interval involved in this study is fairly long, I am wondering if the authors should use the term “temporal stability” (throughout the manuscript). 

A: We used the term “temporal stability” in the abstract and all over the manuscript. However, we sometimes have left the term “test-retest reliability” (or similar expressions) to emphasize the presence of this desirable methodological feature of the scale in a validation study. We think, in any case, that the concepts regarding what we did to establish temporal stability are clear for the reader.

3. Line 50: “largely unrelated to” may be somewhat strong. It might be more accurate to state “only modestly related to”.

A: Done.

4. Line 114, wouldn’t it be better to spell out “Mage” (i.e., “Mean age”)? 

A: Done all over the manuscript.

5. Line 165, it was unclear whether facet scale scores were computed based on raw scores or ipsatized scores. Given that the scales are well balanced in direction, I am wondering if scales scores can simply be computed using raw scores. In any case, the authors may want to make it clear for future users of the inventory whether they recommend using ipsaized or raw scores in computing the scale scores.

A: Consistently with what we have done on the items, the analysis on the facets used the ipsatized data. However, as referred also in the answer to the point 1 of R1 (above), in the conclusions of the paper we have suggested to use the mean scale scores based on raw items.

6. Line 204, the Coefficient alpha for the Unconventionality facet scale have been found to be fairly low from adult samples as well (typically in the .50s). 

A: This is a very interesting and important information that we failed to report in our paper. Now we have added it in the discussion, citing similar Cronbach’s alphas of two recent works on adults (Lee, K., et al., 2018; Henry, S., et al., 2022). Thanks to R2 for this suggestion.

7. I don’t see the need to conduct measurement invariance analyses across three randomly selected subgroups and these analyses could be removed for the sake of simplicity. 

A: We followed the suggestion by R2 and eliminated all over the text the reference to the invariance of the three random subgroups.

8. I think the term, “exploratory” factor analysis is more commonly used than is “explorative” factor analysis. 

A: Done.

---

## [Editor Report · Decision Letter 1]

2 Jan 2023

Development and validation of an instrument to measure personality in adolescence: The HEXACO Medium School Inventory Extended (MSI-E)

PONE-D-22-29099R1

Dear Dr. Gnisci,

We’re pleased to inform you that your manuscript has been judged scientifically suitable for publication and will be formally accepted for publication once it meets all outstanding technical requirements.

Kind regards,

Frantisek Sudzina

Academic Editor

PLOS ONE